# Record-breaking statistics detect islands of cooling in a sea of warming

Elisa T. Sena[1], Ilan Koren[2], Orit Altaratz[2], Alexander B Kostinski[3]

[1]Multidisciplinary Department, Federal University of São Paulo, São Paulo, Brazil
[2]Department of Earth and Planetary Sciences, Weizmann Institute of Sciences, Rehovot, Israel
[2]Department of Physics, Michigan Technological University, Houghton, USA

*Correspondence to*: Ilan Koren (ilan.koren@weizmann.ac.il)

**Abstract.** Record-breaking statistics are combined here with a geographic mode of exploration to introduce a record-breaking map. We examine time series of sea surface temperature (SST) values and show that high SST records have been broken far more frequently than the expected rate for a trend-free random variable (TFRV) over the vast majority of oceans (83% of the grid cells). This, together with the asymmetry between high and low records and their deviation from a TRFV, indicates SST warming over most oceans, obtained using a distribution-independent, robust, and simple-to-use method. The spatial patterns of this warming are coherent and reveal islands of cooling, such as the "cold blob" in the North Atlantic and a surprising elliptical area in the Southern Ocean, near the Ross sea gyre, not previously reported. The method was also applied to evaluate a global climate model (GCM), which reproduced the observed records during the study period. The distribution of records from the GCM pre-industrial (PI) control run samples was similar to the one from a TFRV, suggesting that the contribution of a suitably constrained internal variability to the observed record-breaking trends is negligible. Future forecasts show striking SST trends, with even more frequent high records and less frequent low records.

## 1 Introduction

Man-made greenhouse gas (GHG) emissions have caused staggering changes to the climate system (Masson-Delmotte, 2021), made evident by trends in various types of observations, including: i) an increasing trend in the mean global temperature and ocean heat content (Hansen et al., 2010, Cheng et al., 2017); ii) sea-level rise, mostly due to the thermal expansion of the oceans and, to a lesser extent, due to ice melting over land (Llovel et al., 2019, Watson et al., 2015, Dangendorf et al., 2017), and iii) the elevated frequency and intensity of extreme events, and land and marine heatwaves (Alexander, et al., 2016, Pendergrass and Hartmann, 2014, Myhre et al., 2019, Frölicher et al, 2018, Laufkötter et al., 2020, Oliver et al., 2018). Still, the analysis of climatological time series poses many challenges to quantitative trend extraction because probability distributions of climatological variables are usually not known a priori (Wilcox, 2003, Ghil et al., 2002, Gluhovsky and Agee, 2007), non-linearity effects arise from the complex processes and mechanisms involved, and the entanglement of natural and anthropogenic effects obscures true trends in short time series. Besides these factors, climatological time series are often

composed of datasets measured by different instruments (satellites, for example) or use slightly different measuring methodologies, which may lead to discontinuities, among other problems. All these factors point to the need to pursue robust distribution-invariant ways of extracting trends from intermittently sampled climatological time series, and record-breaking statistics is one such approach (e.g., Anderson and Kostinski, 2011).

Unlike extreme value statistics, which is not concerned with the position of the observations in the sequence, in record-breaking statistics, this information is crucial, rendering trend detection possible. In their seminal 1954 paper, Foster and Stuart used the asymmetry between low and high records to identify trends and variance in time series, using examples of athletics and meteorological data. Since then, this technique, along with various improvements, has been applied to answer a myriad of questions related to the statistical behavior of time series in finance (Wergen et al., 2011), geophysics (Yoder et al., 2010, Van
Aalsburg et al., 2010), climate (Benestad, 2004, Redner and Petersen, 2006), and many other fields (Krug and Jain, 2005, Shcherbakov et al., 2013). Examples of applications of record-breaking statistics in climatological time series include the study of trends, variability, stationarity, and independence in surface temperature (Anderson and Kostinski, 2010; 2011; 2016; Kostinski and Anderson, 2014), rainfall (Lehmann et al., 2015; 2018), and flood events (Vogel et al., 2001).

Sea surface temperature (SST) is one of the most important and well-sampled physical properties of the ocean, describing the important interface between the ocean and the lower atmosphere. Ocean-atmosphere interactions are key processes in the climate system, affecting energy, moisture, and particle fluxes. The ocean stores energy, distributes heat and moisture, and drives weather systems. Long-term measurements of global SST are readily available from climate archives, allowing an analysis of large-scale trends (Huang et al., 2017, Alexander et al., 2018). We use record-breaking statistics to explore local
trends and identify their spatial patterns in 75 years of global SST data (from 1946 to 2020), a variable that has been shown to increase over time (Wuebbles et al., 2017, Hartmann et al., 2013), using linear regression analysis. The use of record-breaking statistics, presented below, has not been applied to SST data analysis, to the best of our knowledge.

To quantify trends via an excess or deficit of record-breaking events, the observed number of records is scaled by the expected
value calculated for a trend-free random variable (TFRV). The asymmetry between the number of high and low records is another metric employed here to assess a trend in the mean value of the time series. The results are discussed for different oceanic regions, and the spatial patterns of record-breaking are examined. The deviation of the observed number of records from the expected value (in units of standard deviation) is used to gauge the significance of the results. Similar analyses can be performed on many other key climatological variables to explore their trends and significance. To gauge the contribution
of internal variability to SST trends and their future projections, record-breaking statistics was applied to the outputs from a global climate model (GCM) from the Coupled Model Intercomparison Project Phase 6 (CMIP6) of the World Climate Research Programme (Eyring et al., 2016).

## 2 Methods

### 2.1 A brief summary of the record-breaking theory

For a given time series of a random variable (x), the likelihood of breaking a record at the $n^{th}$ element is equal to the likelihood that all the preceding values in this time series are smaller than it. Therefore, if the time series is of a trend-free, independently drawn random variable (TFRV), the likelihood of breaking a record decreases monotonically at a rate of 1/n. The first sample is defined to be record-breaking. For a TFRV (arbitrary distribution), the probability that the second sample will be higher than the first is ½, for the $3^{rd}$ to be higher than the first two is 1/3, etc. Accordingly, the probability for the $n^{th}$ variable is:

$$P(n) = \max(x_1, \dots, x_n)) = \frac{1}{n}. \quad (1)$$

Therefore, the expected number of records after n time steps, *H(n)*, is given by:

$$H(n) = \sum_{i=1}^{n} P(i) = 1 + \frac{1}{2} + \frac{1}{3} + \dots + \frac{1}{n}. \quad (2)$$


*H(n)* is the expected number of records over numerous TFRV data vectors of length *n*. Note that *H(n)* increases logarithmically (the slowest possible convergence).

To calculate the variance of *H(n),* we follow Glick (1978) and use an indicator function, replacing the random vector $\boldsymbol{x(n)}$ with a binary function $\boldsymbol{y(n)}$ that is zero everywhere but on $\boldsymbol{x_i}$ samples that break the record of all samples before them. Therefore, the mean of the $\boldsymbol{i^{th}}$ sample, $E(\boldsymbol{y_i}) = \frac{1}{i}$, and the variance can be calculated directly as $V(\boldsymbol{y_i}) = E(\boldsymbol{y_i}^2) - E(\boldsymbol{y_i})^2$, namely $V(\boldsymbol{y_i}) = \frac{1}{i} - \frac{1}{i^2}$. Since there are no correlations between any pair of the $\boldsymbol{y_i}$ -s, the sum of $\boldsymbol{y_i}$ that measures the average expectations for the number of $\boldsymbol{x_n}$ breaking is, as expected, $\boldsymbol{H(n)} = \sum_{i=1}^{n} \frac{1}{i}$ and the variance $\boldsymbol{V(n)} = \boldsymbol{V(x_n)}$ is simply the sum

of all $\boldsymbol{y_i}$ variances:

$$V(n) = \sum_{i=1}^{n} \frac{1}{i} - \sum_{i=1}^{n} \frac{1}{i^2} \quad (3)$$

By knowing the expected mean and variance for the TFRV number of record-breaking events, one can estimate the magnitude and significance of the deviations from them.

In this work, we introduce 'record-breaking maps' that provide information on the geographical structure of SST trends. The advantages of this method are its simplicity and generality. Since only serial ordering matters, the method handles non-uniform or intermittent sampling and it is not limited to linear trends. It is robust insofar as it is distribution-invariant (that is, it is non-parametric, since it does not depend on the distribution of the underlying random variable) and unit-insensitive (independent

of magnitude). Moreover, when applied together with spatial information of many adjacent pixels (as is done in this work),
such analysis easily detects coherent spatial structures of trends and their significance. This can be useful for analyzing the
statistics of climate variables.

## 2.2 Significance assessment with the Kolmogorov-Smirnov (KS) test

The complete distribution of records needed as a reference in the KS test, is a function of all possible permutations for record-
breaking sequences. Arnold et al. (2011) have shown that the exact theoretical probability of a TFRV to break k records in a
time series of length n is given by the unsigned Stirling number of the first kind divided by the factorial of the time series
length:

$$P(k) = \frac{|S(n,k)|}{n!}, \quad (4)$$


where $|S(n,k)|$ are the coefficients of $x^k$ in a rising factorial of $n$ terms, that is,

$$\prod_{i=1}^{n}(x + i - 1) = \sum_{k=0}^{n}|S(n,k)|x^k. \quad (5)$$

A complete description of the Stirling numbers and how to calculate them using a recurrence relation can be found in e.g.,
Jordan (1950).

The collective significance of SST trends over the entire globe was assessed by the one-sample non-parametric KS test. For
that, the cumulative distribution function (CDF) of the observed number of records was compared with the TFRV theoretical
distribution from Eq. (4). The maximum distance D between the CDFs was used to infer the significance of the observed
records. We compared the observed D values with the distribution of D values obtained by applying the KS test to 3000 CDFs
obtained from three million Monte Carlo simulations of TFRV records, 75 samples long each. A detailed explanation of how
to calculate the significance using the KS test is provided in Glienke et al. (2020).

## 2.3 Data

The method is applied to 3D datasets in which the horizontal x and y indices represent the location (longitude and latitude),
and the third dimension is the time of the sea surface temperature (SST) measurement. We treat each pixel as an independent
time series. The dataset used in the analyses comprises the monthly global estimates of SST from NOAA's extended
reconstructed sea surface temperature, version 5 (NOAA-ERSST-v5), from 1946 to 2020 (Huang et al., 2017). It is a
combination of in situ observations from ships, buoys, and Argo floats above 5 meters, and the Hadley Centre version 2 ice-
SST concentration (HadISST-v2) derived from the International Comprehensive Ocean-Atmosphere Dataset (ICOADS)

Release 3.0. Based on monthly error estimates (Huang et al., 2016), the SST annual mean error was set at 0.1ºC. To avoid many ties, pixels covered by more than 90% of sea ice were excluded from the analysis, as in this case, the SST is set to the freezing point temperature of -1.8ºC. Although this dataset provides SST estimates since 1854 in a 2º x 2º grid resolution, the analysis was from 1946 to the present to avoid biases and artifacts due to bucket design, different measurement approaches,

and data truncation that resulted in spurious SST measurements, especially by Japanese and German ships before the end of World War II (Chan et al., 2019). This dataset is being continuously improved and current and previous versions have already been applied to infer SST trends (e.g., Deser et al., 2010, Wuebbles et al., 2017, Chan et al., 2019).

## 2.4 Global climate model

Here we argue that record-breaking statistics can function as a diagnostic tool when examining performance of global climate
models (GCM) and, in particular, to infer the contribution of the internal climate variability to SST trends and to estimate future projections. An extensive analysis of the number of records was performed with the Meteorological Research Institute Earth System Model version 2.0 (MRI-ESM2-0) (Yukimoto et al., 2019), aiming to compare SST trends from observations, TFRV, GCM pre-industrial (PI) GHG runs, GCM during the study period and future GCM projections. This model was chosen because it has a transient climate response (TCR) of 1.6 K, similar to observations (Meehl et al., 2020). Also, it predicts
moderate increases in temperature and presents low residues, relative to other CMIP6 models (Zelinka et al., 2020 – supplementary material). The methodology was applied to global time series of yearly-averaged SST outputs from the GCM, in 1º x 1º spatial resolution, simulated in different scenarios.

The contribution of natural factors to the SST trends was assessed using the pre-industrial control run (PI control) from the model, which simulates a scenario without an increase in GHG after 1850, therefore accounting only for internal variability of
the climate system. The distribution of the number of records was assessed from 1000 random 75-year samples of global simulations from the PI control run of MRI-ESM2-0 model. Ties were broken by adding a small random noise to each SST value. Record-breaking statistics was also applied to the outputs of the GCM simulations of the shared socioeconomic pathway 2 (SSP2-4.5), which considers the intermediate GHG emissions scenario, and during the same time period as used in our observational analysis (using a combination of the historical data set and SSP2-4.5).

## 3 Results

The record-breaking maps of the observed number of high and low records of SST for the 75-years of the study period are presented in Fig. 1a,b. To illustrate the procedure, we chose time series of annual mean SST for two pixels located in points A (0ºN; 95ºE) and B (40ºN; 195ºE) (panels 1c and 1d). High and low records of SST are marked with red and blue dots, respectively. There is an increasing trend at point A, with 15 high records and 3 low records. In contrast, point B series presents
more low than high records, i.e., 7 vs. 4, respectively. This asymmetry is further explored by using the number of records to gauge trends in the SST and identify their spatial patterns.

Some simple metrics proposed by Foster and Stuart (1954) and Anderson and Kostinski (2011), may be used as indicators of possible trends in the time series of highly variable data. One such metric is obtained simply by dividing the number of observed records, $R(n)$, by the expected number of records for a TFRV, $H(n)$. The ratio $R(n)/H(n)$ can be used to quantify by

how much the observed number of records exceeds or falls short of the TFRV expectation for both the high and low records. In addition to the previous indicators, the asymmetry between the number of high and low records can also be used as a metric for the time series trend. Here, we use the natural logarithm of the ratio between $R_{High}$ and $R_{Low}$ as a trend indicator to complement the previous analyses (Anderson and Kostinski, 2011),

$$\rho = ln\left(\frac{R_{High}}{R_{Low}}\right), \qquad (6)$$

where sign of $\rho$ distinguishes between increasing and decreasing trends, and its magnitude is related to the slope of the curve. These tests explore different aspects of the record-breaking statistics and offer additional support for the presence of a trend in noisy data.

The record-breaking maps of the observed number of high and low SST records over the TFRV expected number of records
($R(n)/H(n)$), the trend indicator index $\rho$ (Eq. 6), and their frequency distributions are shown in Fig. 2. The red (blue) color in the maps indicates excess (deficit) of high records in panel a, deficit (excess) of low records in panel c, and positive (negative) values of $\rho$ in panel e. The contrast between the distributions of the high and low records is evident. The expected number of records for a TFRV time series of length 75 is ~5. As Figs. 1 and 2 demonstrate, for most of the global oceans, the observed number of high records far exceeds the TFRV expected value of 5, while for the low records, it is below 5 in most pixels. In
83% of the grid cells, the number of high records is above the expected, and in 17% it is more than twice the expected value for a TFRV. It is particularly evident in the tropical and subtropical Atlantic, the Central Pacific, the Eastern Indian Ocean, and the Southern sea. Conversely, the number of low records seldom exceeds the TFRV expected value by more than twice (less than 0.1% of the pixels), and $R_{Low}(n)/H(n)$ is below 1 in more than 72% of the grid cells. Fig. 2c depicts the trend indicator index $\rho$. Positive values of $\rho$ are observed over most of the globe. The ratio between high and low records is higher than 1 in
88% of the globe, and higher than 2 in 51% of the pixels. This asymmetry in the number of high and low records is a good indicator of a trend in the SST. It is interesting to notice that, even though the overwhelming majority of the oceanic area is warming (as evidenced by the predominance of the red color in all record-breaking maps), there are some islands of consistent cooling (where the spatial patterns show a coherent persistence of the color blue in many neighboring pixels). We will discuss some of the most interesting cooling islands' features in the next section.

We also examined differences in regional trends (the regions are marked in Fig. 1b). Fig. 3 presents boxplots of the distribution of high and low SST records over the expected TFRV value ($R(n)/H(n)$) and of $\rho$ for each region. Once again, the results confirm the asymmetry between the number of high and low records. The global distribution of high records is shifted to the right (towards higher values) compared to the expected distribution of records for a TFRV. All regions show an excess of high records compared to both the expected value for a TFRV and to the number of low records. The asymmetry between the number

of high and low records is evident in all oceanic regions since the medians of the distribution of the observed high (low) records were higher (lower) than 1 in all of them (Figs. 3a and 3b). The Arctic, Indian, South Pacific, South and North Atlantic Oceans are the regions where the excess of high records is more prominent, all of them exhibiting values of over 75% above the expected number (Fig. 3a). The South Atlantic Ocean maintains the highest ratio between high and low records (Fig. 3c). These results further confirm the global extent of the positive trend in SST; global warming of the upper oceans.

Are the number of records observed in the global SST 75-year time series significantly different from those calculated for a TFRV time series of the same length? To answer this question, Fig. 4 compares the cumulative distribution of the observed number of high (red) and low (blue) records of all the pixels (over the whole globe) with the theoretical TFRV distribution of records (green). It is clear that the number of high records is always larger than the one for the theoretical distribution; that is, the observed curve is shifted to the right compared to the theoretical one. The opposite behavior is observed for the number of

low records. The Kolmogorov-Smirnov (KS) test indicates that the maximum distance D between the distribution of observed high (low) records and the theoretical cumulative distribution function (CDF) of the number of records expected for a TRFV (Eq. 3) is DH=0.32 (DL=0.31). To assess the significance of the D values obtained for the observed records, the KS test was applied on 3000 CDFs of the number of records obtained for TFRV Monte Carlo simulations. The probability distribution function (PDF) of the maximum distance D for the Monte Carlo simulations is shown in Fig. 4c. The green dashed line shows

the 99.9% percentile ($D_{99.9}$), and the maximum deviations, DH and DL, are highlighted in red and blue dashed lines, respectively. The results show that the observed DH and DL are overwhelmingly significant, both about 6 times $D_{99.9}$, the distance expected for the 99.9% percentile of the Monte Carlo simulations' PDF. Thus, the global ocean SST does not represent a TFRV, and warming is evident.

  Having established the global (bulk) significance of SST trends, we next consider the local significance in order to explore

spatial patterns. To that end, for each 2º by 2º pixel of the grid, we analyzed the deviation from expectation in standard deviation units, given by the difference between the observed number of records, R(n), and the expected number of records, H(n), for a TFRV, divided by the theoretical standard deviation, $\boldsymbol{\sigma} = \sqrt{\mathbf{V(n)}}$, where $V(n)$ is the variance of the TFRV distribution (Eq. 3),

$$\boldsymbol{Dev} = \frac{\mathbf{R(n)-H(n)}}{\boldsymbol{\sigma}} . \quad (7)$$

The record-breaking maps in Fig. 5 show that in 15.8% of the pixels, the number of high records exceeds the expected TFRV value by more than 2 standard deviations, about 12 times as many pixels as for TFRV. In about 2.4% of the pixels, it is above 3 standard deviations, in contrast to a TFRV value of (virtually) zero. We note the clear positive record-breaking trend over almost the entire Atlantic Ocean (except for high latitudes in the northern hemisphere), the tropical Pacific Ocean, and most of the Indian Ocean (especially near India and Western Australia). These positive regional anomalies are more comprehensively

addressed below.

  The low records show large deviations from TFRV in the Arctic region, in the North and South Atlantic, and in some regions of the Indian and Pacific Oceans (where the colors are dark red, signalling fewer records than expected). The histograms in

panels 5c and 5d show ***Dev*** values as high as 6 for high records and 3 for low records. Although the associated theoretical PDF of the number of records for a TFRV is asymmetric, the difference between the mean and the median of the distribution

for 75 observations is very small (4.9 and 5, respectively). Therefore, using the normal approximation, the deviations displayed in Fig. 3 are again highly significant and demonstrate that the number of records of the SST time series of each pixel is far from that calculated for a TFRV. For example, a 2-standard-deviation value would be very unlikely for the theoretical distribution of the records of a TFRV time series, being above the 95th percentile.

To infer the contribution of natural variability to the number of records, as well as to verify future projections, record-breaking statistics was applied to GCM SST outputs in three different conditions: i) PI control run samples, ii) during the study period (1946-2020) and, iii) for SSP2-4.5 scenario projection (from 2015-2100). Figure 6 shows boxplots of the high and low number of records over the TFRV expected value (***R(n)/H(n)***) and $\rho$ for the observational dataset, Monte Carlo simulations of a TFRV, and modelled SST from the GCM. Recall that boxplots are designed to allow the comparison of the distributions at a glance.

The coloured box is delimited by the first and third quartiles, The thick line inside the box shows the median for each distribution. The size of the whiskers is 1.5 the interquartile range. Therefore, any point outside the whiskers is considered an outlier.

The boxplots show that the distribution of the metrics of interest (***R(n)/H(n)*** and $\rho$) of the GCM PI control realizations are almost identical to that of a TFRV, in contrast to the distribution of the observational dataset. Note that although our reference

is named a trend-free random variable (TFRV), it is devoid of trends not only in the mean value but in variance and other moments as well. In other words, it is an independent, identically distribution (I.I.D.) random variable. Thus, we expect the climate internal variability to affect record highs and lows and exceed I.I.D variance. Yet, the expected standard deviation of the records of a TFRV is 1.8, and the model's PI control realizations have a standard deviation of 1.9. This similarity of the records distribution drawn from the GCM PI control with the TFRV gives us more confidence in applying record-breaking

statistics to verify trends in climate time series and, once again, confirms the robustness of this method.

On the other hand, the observed, modelled and projected ***R(n)/H(n)*** and $\rho$ are quite different from that of a TFRV, clearly shifted to higher (lower) values in case of high (low) records. Note that the GCM during the study period could capture the general trend revealed by record-breaking statistics, but smoothen its magnitude. Future projections, however, are outstanding, with positive values of $\rho$ in more 95% of the global oceanic area. predicting that in the future the high record breaking will be

even more frequent than in the present, while low record breaking will be even less frequent. It was also very interesting to test similarity of the spatial patterns of this global model simulation and the observation-based results during the study period (see the Supplement). Boxplots of the deviation from expectancy (in units of standard deviation) reinforce the previous findings (Figure 7). Both, the distribution of the deviation from expectancy of the PI control run samples and the TFRV are centered at zero, in contrast to the results for the observational dataset and GCM historical and future simulations (which were clearly

shifted to positive values in the case of high records and negative values for low records), confirming the significance of the results.

**4 Discussion**

This work uses record-breaking statistics for identifying trends in a global dataset of SST time series (between 1946 and 2020) and explores their spatial patterns. This method has many advantages such as being free from assumptions on the distribution.
The record-breaking maps allow for both estimating the existence and significance of trends locally (per pixel, see Figs. 2 and 5 as examples) and detecting spatial structures (such as areas with significantly strong trends or, in contrast, areas with opposite trends). These spatial patterns can hint at the geophysical processes controlling the explored variable. This method is particularly well-suited for "many but short" datasets consisting of numerous relatively short time series, as is often the case in climatology, and is satellite meteorology, in particular.

The asymmetry between the numbers of high and low records and their deviation from the theoretical calculation for a TRFV indicate a significant increasing trend in SST over most of the Oceans but also expose some cooling regions (Figs. 2-5). Comparing panels 2a and 2c reveals extensive warming regions, associated with large numbers of high records and small numbers of low records, in the tropical and mid-latitude Northern Atlantic Ocean, the whole South Atlantic Ocean, south of
Australia, in the vicinity of Indonesia (in the Indian Ocean), and near the west coast of Mexico (in the Pacific Ocean). Despite the overwhelming warming, coherent cooling patterns are also observed in regions characterized by a small number of high records (Fig. 2a) and a large number of low records (Fig. 2c). Some of the regions where low records were broken more frequently (like in the tropical eastern Pacific and in the Indian Ocean) may be associated with strong events of the negative phase of natural climate oscillations like El Niño Southern Oscillation (ENSO) and the Indian Ocean Dipole (IOD) in the
Southern Hemisphere. Previous observations that showed cooling in part of the North Atlantic Ocean followed by warming in the last two decades (Deser et al., 2010) are consistent with the maps in Fig. 2. Also noteworthy are the large spatial clusters that show the same trend signals, proving the consistency in the spatial correlations of $R(n)/H(n)$. Next, we focus on some interesting islands of cooling, such as the high latitudes of the Northern Atlantic, near Greenland, some areas in the North Pacific, and, even more interestingly, in the Southern Ocean between Australia and South America.
The cooling observed here in the North Atlantic region is consistent with a phenomenon known as "the cold blob", a region in the ocean near Greenland that has been experiencing cooling over the years despite global warming. The causes for this cooling have been associated with ice melting in high latitudes that induces changes in the ocean circulation due to increased freshwater fluxes, causing a slowdown of the Atlantic Meridional Overturning Circulation (AMOC) and reduced heat transport to that region (Josey, et al., 2018, Rahmstorf et al., 2015). More recently, other factors, such as heat transport out of the "cold blob"
to higher latitudes and increased cloudiness over the area, have also been linked to the cooling in this region (Keil et al., 2020). The causes of such cooling have been linked to human footprint, mainly the anthropogenic release of greenhouse gases into

the atmosphere, as opposed to natural variability (Chemke et al., 2020). A similar cooling pattern is observed here near Antarctica, around longitude 200°. Cooling in the surface of the Southern Ocean and warming of the subsurface have been previously reported (Masson-Delmotte, 2021, Haumann et al., 2020). However, unlike previous findings, in the present analysis, the cooling trend seems to appear in a very specific geographical spot, i.e., near the Ross gyre (Fig. 2). The mechanisms associated with such cooling are still under investigation. Upwelling delays the warming of the sea surface but cannot explain the cooling. Haumann et al. (2020) proposed that the cooling is due to wind-driven sea-ice transport and its subsequent melting, decreasing the salinity of the upper ocean. Freshening of the upper ocean weakens convection and vertical mixing, preventing the warming of the ocean surface (Armour et al., 2016). Finally, the cooling pattern in the North Pacific is not as consistent as the other ones. The time series in this region presents an SST increase until around the 1980s and then a decrease. Analyzing SST from observations and models, Deser et al. (2010) found a low significance of trends in this region. Record-breaking statistics was also applied to a GCM to explore the contribution of internal variability to climate trends, the reliability of historical simulations and future projections of records. The similarity between the distribution of the number of records from the model's pre-industrial control run and that of a TFRV, opposed to the observational SST, is noteworthy, This result gives us confidence that the positive and negative trend signals observed in Figures 1 to 5 cannot be explained by natural variability alone. Therefore, the "sea of warming" and "islands of cooling" hereby detected are the result of the complex interactions of the climate system, when forced by human-induced climate change. The results have also shown that the GCM was able to fairly reproduce the observed records during the study period (Figures 6 and 7), including the general warming and cooling spots observed in the spatial patterns of records (see Figure S6 of the supplementary material). When applied to future GCM projections of SST, record-breaking statistics shows strong positive trends, with the number of high records surpassing the number of low records in the overwhelming majority (more than 95% of the pixels) of the globe (see also Figure S7 of the supplementary material). The analysis performed here shows that record-breaking statistics is a powerful tool capable of detecting trends in noisy global time series and demonstrates how to evaluate their significance in climate science studies.

**Acknowledgment**

This project has received funding from the European Research Council (ERC), under the European Union's Horizon 2020 research and innovation programme (CloudCT, grant agreement No 810370). E. T. S. also acknowledges CNPq Universal Project grant 421870/2018-4, and A. B. K. acknowledges NSF grant AGS-1639868 and 2217182. We thank NOAA/OAR/ESRL PSL, Boulder, Colorado, USA, for processing and maintaining the NOAA_ERSST_V5 data in their website. We also thank the World Climate Research Programme, which, through its Working Group on Coupled Modelling, coordinated and promoted CMIP6, as well as the climate modeling groups for producing and making available their model output, the Earth System Grid Federation (ESGF) for archiving the data and providing access, and the multiple funding agencies who support CMIP6 and ESGF.

**Code and data availability**

All the datasets used in this study are publicly available and can be downloaded from their respective websites. All analysis codes were specifically developed for this study and are available from the contact author upon request.

**Author contributions**

ETS, IK, OA and AK designed and performed the research, analyzed the data, and wrote the paper.


**Competing interests**

The contact author has declared that none of the authors has any competing interests.

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

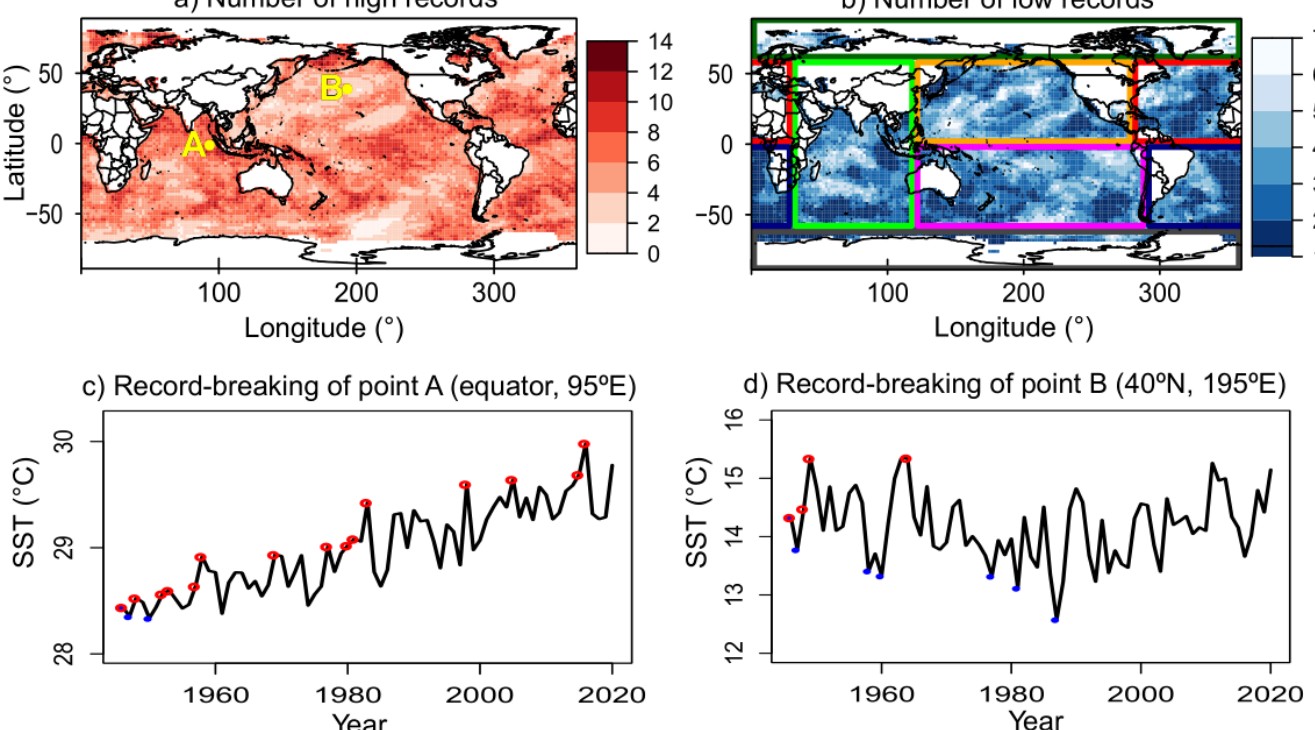

**Figure 1:** Number of observed high (a) and low (b) records of SST for the p 1946 to 2020 period. Panels (c) and (d) show samples of time series of annual SST for two pixels, marked by yellow dots on panel (a): points A (0ºN; 95ºE) and B (40ºN;

195ºE), high/low records are marked by red/blue dots on the time series. Location A shows a clear increasing trend, with high
records exceeding the low ones, while at location B, the opposite holds. The asymmetry between the number of high and low
records and their deviation from the expected value for a TFRV were used to quantify trends and their significance in each
pixel's time series. The boxes in (b) divide the global oceans for subsequent analysis: Antarctic (ANT - gray), Arctic (ARC -
dark green), Indian (IND - green), North Pacific (NP - orange), South Pacific (SP - magenta), North Atlantic (NA - red), and
South Atlantic (SA - navy blue).


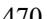

**Figure 2:** Number of observed high (a) and low (c) records of SST over the expected value for a TFRV in a 75-years time
series. The trend indicator $\rho$, quantifies the asymmetry between the number of high and low records, $\rho = \ln(R_{High}/R_{Low})$ (e),
and their respective distributions for the whole globe (b, d, f) are presented. The red dashed lines mark the thresholds where
observed records exceed the expected value in (b) and (d) or when the number of high records is larger than the low records
in (f). These three different metrics combined can be used to simply and robustly detect trends in noisy time series. The

asymmetry between high and low records is evident, with a large number of high records and a small number of low records over most of the globe, particularly in the tropical and subtropical Atlantic, the Central Pacific, the eastern Indian Ocean, and the Southern sea. Note that, although the overwhelming majority of the ocean is warming, some islands of persistent cooling are observed in the maps.

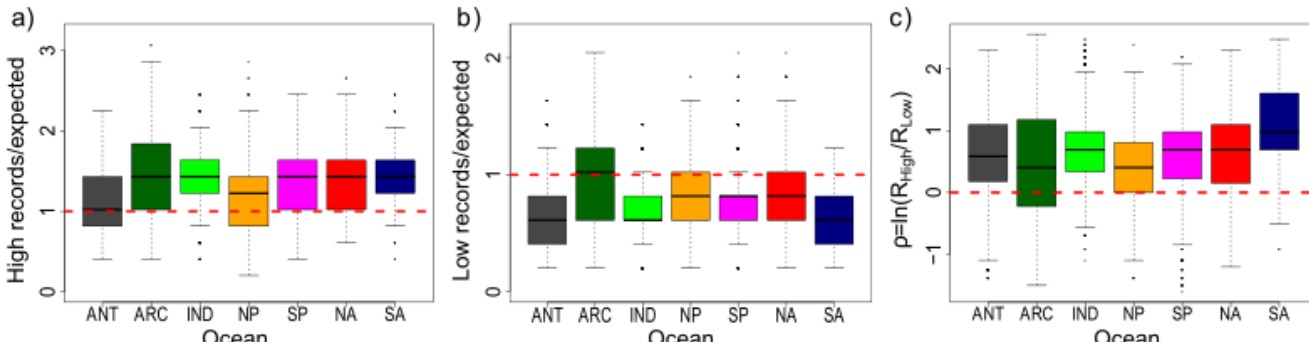

**Figure 3:** Boxplots of the number of high (a) and low (b) records over the expected value for a TFRV, and $\rho = \ln(\text{high/low})$ records (c) for different oceanic regions (as marked in Fig. 1): Antarctic (ANT - gray), Arctic (ARC - dark green), Indian (IND - green), North Pacific (NP - orange), South Pacific (SP - magenta), North Atlantic (NA - red), and South Atlantic (SA - navy blue). The red dashed lines in (a,b) mark a ratio of 1 between the observed number of records and the expected value for a TFRV, and in (c) when the high records number is larger than the number of low records. An excess of high records compared to both the expected value for a TFRV and to low records is observed for all regions. The South Atlantic Ocean exhibits the largest ratio between high and low records.

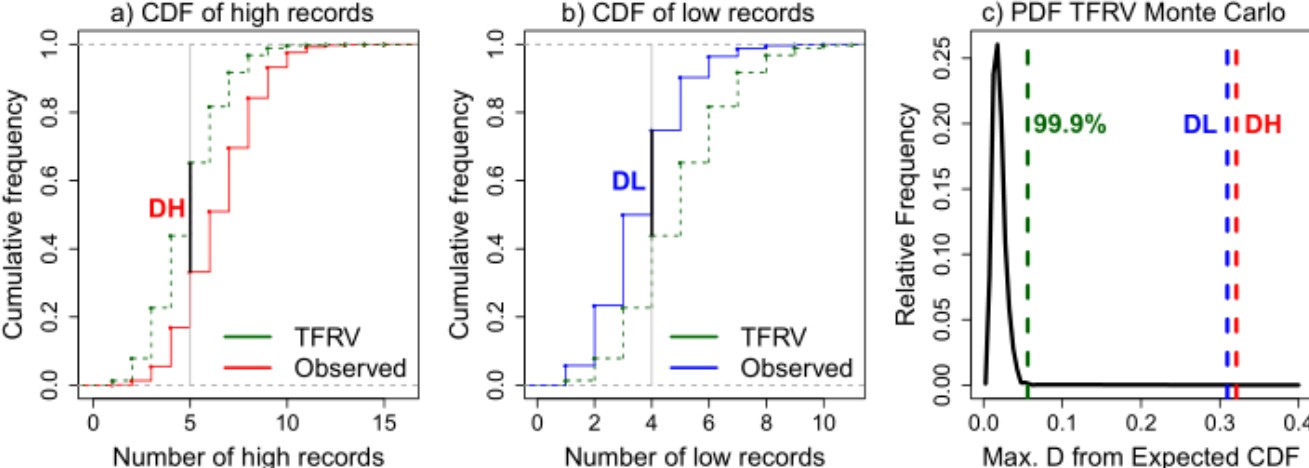

**Figure 4:** Cumulative distribution functions (CDFs) of the theoretical TFRV number of records from Eq. (4) (in green) and number of observed high (red) (a), and low (blue) (b) records. The Kolmogorov-Smirnov test was applied to the distributions, and the maximum distances, DH and DL, were obtained for the high and low records, respectively (black vertical lines). Note that the exact theoretical (discrete and integer-valued) distribution of the number of records of a TFRV time series of length 75, given by Eq. (4) was used. The probability distribution function of the maximum distance D for Monte Carlo simulations

of the number of records of TFRV (c) was used to assess the significance of the results. The 99.9% percentile and the maximum deviations, DH and DL, are shown in green, red, and blue dashed lines, respectively. Both DH and DL far exceed the 99.9%, demonstrating the high statistical significance of SST global trends in high and low records.


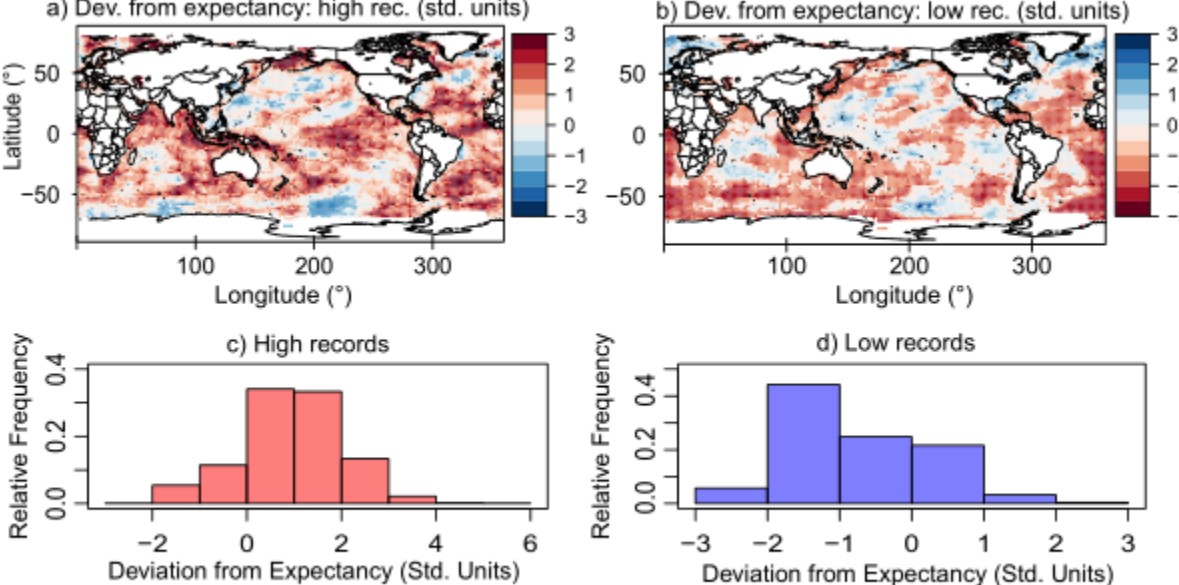

**Figure 5:** Departure of the observed number of high (a) and low (b) records from expectation, in units of standard deviation, and their frequency distributions (c and d). In 15.8% of the pixels, the number of high records exceeds 2 standard deviations

above expectation. This value is ~12 times larger than TFRV. Large deviations are observed over all the oceans for high records and predominantly in the Arctic and Atlantic Oceans for low records.

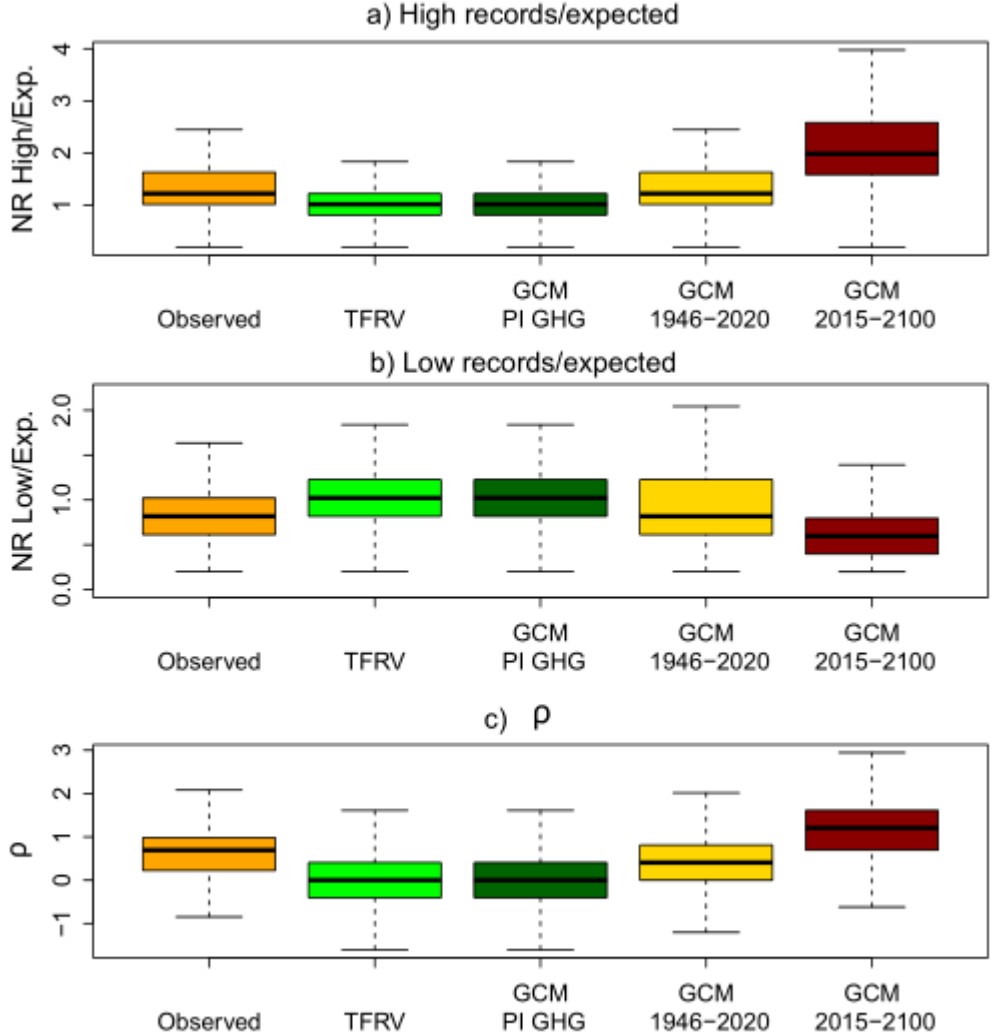

**Figure 6:** Boxplots of the ratio of the number of high and low records over the expected value and of $\rho$ for the observed SST between 1946-2020, Monte Carlo simulation of a TRFV, modelled SST of the PI control run samples, modelled SST between 1946-2020, projected SST between 2015-2100 (using the SSP2-4.5 scenario). The simulations were from the MRI-ESM2-0 CMIP6 global climate model.

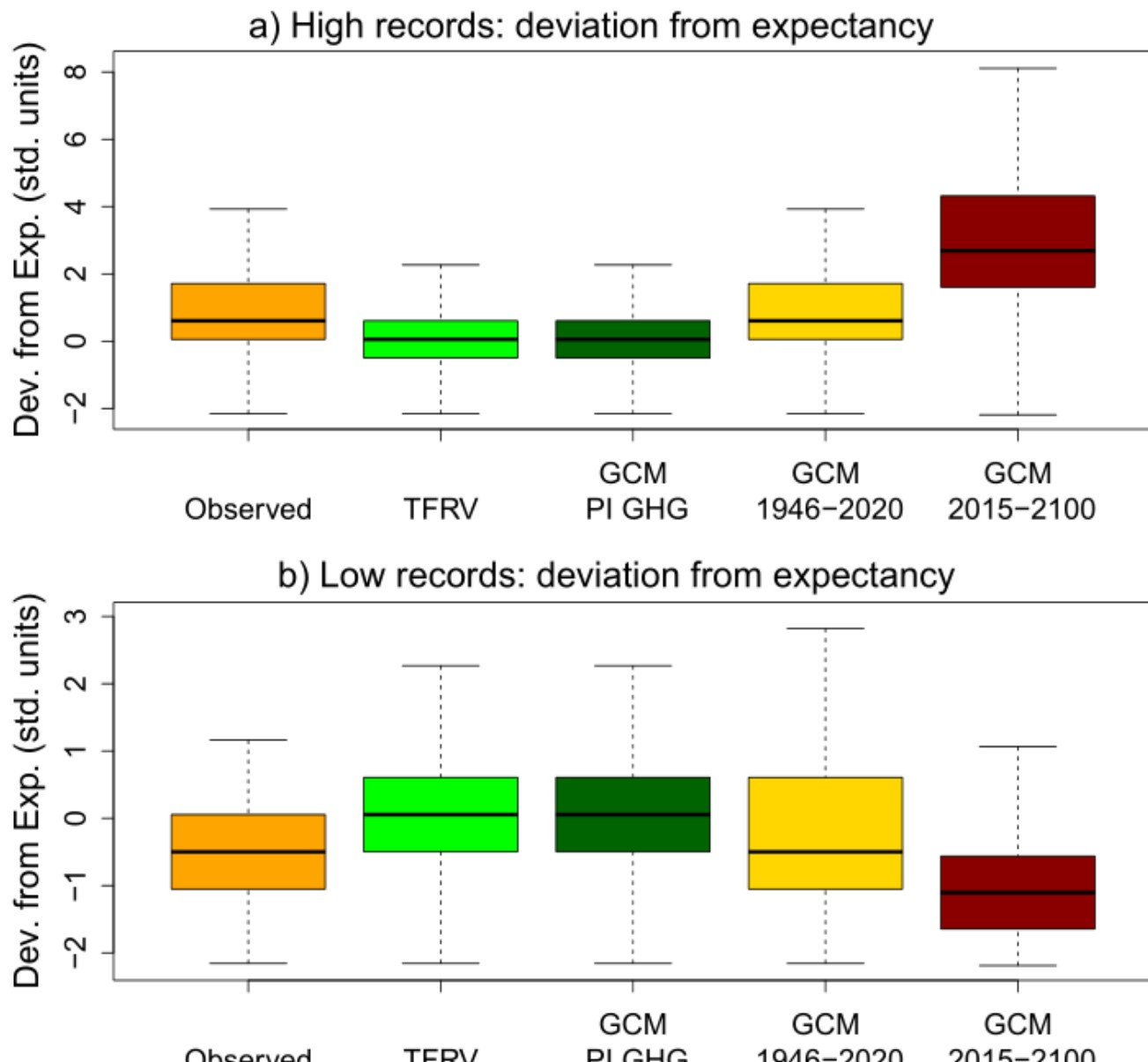


**Figure 7:** Boxplots of the high and low deviation from expected number for: observed SST between 1946-2020, TRFV, modelled SST of the PI control run samples, modelled SST between 1946-2020, projected SST between 2015-2100 (using the SSP2-4.5 scenario). The simulations were from the MRI-ESM2-0 CMIP6 global climate model.