# Peer review of "Record-breaking statistics detect islands of cooling in a sea of warming"

_Atmospheric Chemistry and Physics, 2022_

## Referee Comment (RC1)

In this study, the authors investigate the observed record-breaking SST events by comparing with the expected rate for a trend-free random variable (TFRV). The authors find the asymmetric nature of the high and low records and reveal islands of cooling in the North Atlantic and Southern Ocean. The record-breaking theory is interesting; however, the assumption of the theory may not be enough reliable and the results may be sensitive to the length of a time series. Given these issues, I would recommend that this paper is not suitable for publication in this high-rank journal. More specific comments are listed as below.

Major comments:

1. The results of this study depend mainly on the comparison with the TFRV model. However, the climate system is clearly not a TFRV. Significant trends in SST can be detected even without the influence of human-induced greenhouse gases. As Deser et al.(2013) and Wallace et al. (2015), the internal variability is important for multi-decadal trends in climate variables, which are independent of human activities.

The authors use only a trend-free model for comparison, which is more of a hypothesis testing tool to test observed trends. and is of little implication for the climate community to understand the observations and climate change.

It is suggested that considering internal trends of the climate variables, such as add the trend distribution of Pre-industrial experiments from CMIP5/6 into the record-breaking statistics and comparing it with observed data, may yield more valuable results.

2. The results may depend to a large extent on the sample size. As shown in Equation 2, the broken k-record varies with the length of the time series. In other words, the results may be sensitive to the length of the time series and not robust.

---

## Author Comment (AC1)

**Record-breaking statistics detect islands of cooling in a sea of warming**

**Answers to reviewer 1**

"In this study, the authors investigate the observed record-breaking SST events by comparing with the expected rate for a trend-free random variable (TFRV). The authors find the asymmetric nature of the high and low records and reveal islands of cooling in the North Atlantic and Southern Ocean."

We thank the reviewer for their comments and the time spent with this review. But we must point out that the findings far exceed what is selected above. Below is a list of the findings reported in the original manuscript:

i) A new tool: maps of the number of high and low records of SST based on 75 years of global observations are introduced in Figure 1.

ii) The global spatial distribution of the number of observed high and low records of SST over the expected number of records for a trend-free random variable (TFRV), showing that in 83% of the grid cells, the number of high records is above the expected, and in 17% it is more than twice the expected value for a TFRV, while the number of low records very rarely exceed the expected value for a TFRV (Figures 2a, 2b, 3a and 3b). This is an overwhelming evidence for ocean warming from a novel perspective.

iii) The global spatial distribution of the ratio between the number of high over low records of SST, showing that the ratio of high to low records is exceeds unity in 88% of the globe, and higher than 2 in 51% of the pixels, again pointing to robust trends in the SST time series (Figures 2c and 3c).

iv) The significance of the trends for each pixel was estimated using global maps of the deviation of number of records from expectation in standard deviation units, showing that in 15.8% of the pixels, the number of high records exceeds the expected TFRV value by more than 2 standard deviations (Figure 5).

v) Another important result of this study is the analysis of the collective trends in sets of pixels. The maps of the trends display spatial coherence while the bulk significance of the trends, quantified by the Kolmogorov-Smirnov test, is well above 99.9% (Figure 4c).

vi) Without any "tuning", the approach reveals islands of cooling, such as the well-known "cold blob" in the North Atlantic and a surprising result: a coherent cooling area in the Southern Ocean, near the Ross sea gyre, not previously reported.

The reviewers comments helped us to clarify the above points.

"The record-breaking theory is interesting; however, the assumption of the theory may not be enough reliable and the results may be sensitive to the length of a time series. "

We are glad that the reviewer finds the theory interesting. We must add, however, that contrary to their statement, record-breaking is more robust than conventional methods, commonly used to estimate trends in time series. Specifically, unlike most regression methods, record-breaking statistics is distribution-invariant (that is, it does not depend on the distribution of the underlying random variable). Moreover, it is unit-insensitive, handles

non-uniform or intermittent sampling, and can also be used to identify non-linear trends. When applied in tandem with spatial information of many adjacent pixels (as is done in this work for the 1st time, to the best of our knowledge), such analyses detect coherent spatial structures of trends and their significance.

Returning to the reviewer's specific point: sensitivity to a length of time series, we offer the following technical remarks.

For a TFRV, the probability of breaking a record decreases monotonically at a rate of 1/n. Therefore, the expected number of records in a TFRV is given by Equation 2, that is,

$$H(n) = \sum_{i=1}^{n} P(i) = 1 + \frac{1}{2} + \frac{1}{3} + \ldots + \frac{1}{n}.$$

Hence as time passes (n increases), it becomes harder to break a record. This is the crucial property that implies that record-breaking statistics is actually less sensitive to time series length than other regression methods. For example, consider the following: for a time series of a 1000 years (points), the expected number of records (NR) for a TFRV is about 7. Meanwhile, in our 75-year time series (expected NR = 5) there are many pixels with NR > 8 and even reaching NR=14.  Hence, n=75 is "as good as" n=1000.  Thus, this is a conclusion hardly "sensitive to a length of a time series", and unmatched by the conventional regression methods.

We revised the manuscript to clarify this point by emphasizing the robustness of the record-breaking statistics its insensitivity to time series length. We also applied record-breaking statistics to CMIP6 GCM pre-industrial control runs and compared them to the expected TFRV results, as suggested by the reviewer (see below). The main conclusion from these analyses was that the distribution of records from GCM PI control run samples is very similar to the one from a TFRV. The revised manuscript includes the main results (Figures S2 and S6 below) and conclusions from this analysis. The focus of this paper, however, remains on identifying trends in SST observations with a robust method.

Major comments:

1. "The results of this study depend mainly on the comparison with the TFRV model. However, the climate system is clearly not a TFRV. Significant trends in SST can be detected even without the influence of human-induced greenhouse gases. As Deser et al.(2013) and Wallace et al. (2015), the internal variability is important for multi-decadal trends in climate variables, which are independent of human activities."

As explained above, no claim is made in the paper that climate system is a TFRV.  We use TFRV merely as a reference which, in contrast to other methods, is distribution-independent, e.g., whether the fluctuations are uniform, normal, lognormal, etc.

As for the reviewer's second point: the goal of this study was only to identify trends in SST time series and discuss their significance using a robust method, rather than to separate the anthropogenic influence from the internal natural variability of the climate system. These points are amplified in the revised  manuscript.

"The authors use only a trend-free model for comparison, which is more of a hypothesis testing tool to test observed trends and is of little implication for the climate community to understand the observations and climate change.

It is suggested that considering internal trends of the climate variables, such as add the trend distribution of Pre-industrial experiments from CMIP5/6 into the record-breaking statistics and comparing it with observed data, may yield more valuable results."

We thank the reviewer for the suggestion. Our study did not aim to separate human-induced warming from natural variability and, therefore, in the original manuscript, pre-industrial experiments from CMIP5/6 were not considered. Rather, our aim was to identify observed trends. Therefore we used one of the best-sampled, high-quality data set of SST observations. However, as pointed out by the reviewer, record-breaking statistics can be applied to global climate models to explore the contribution of the internal climate variability to SST trends.

Following the reviewer's comments, we performed a thorough analysis aiming to compare SST trends from observations, TFRV and GCM pre-industrial (PI) GHG runs. We have analyzed MRI-ESM2-0 CMIP6 global climate model (GCM). This model was chosen because, according to Meehl et al., 2020, it has a transient climate response (TCR) of 1.6 K (midrange suggested by the reviewer 2), and because it predicts moderate increases in temperature, relative to other CMIP6 models and presents low residues (Zelinka et al., 2020 – supplementary material).

Record breaking statistics was applied to global time series of simulated SST (1 x 1 degree lat/lon) in the **pre-industrial control run** (PI control) from the model, which keeps the GHG levels steady after 1850 (considered the reference year for the transition from the pre-industrial era to the present). 701 years of PI control run were available for this model. As suggested by reviewer 2, for each grid cell we took 1000 random 75-year chunks from the PI control global simulation of MRI-ESM2-0 model to explore the effect of natural variability in the number of records. Ties were dealt with by adding a small random noise to each SST value.

We compared the number of records of the modelled PI control (which accounts only for internal variability of the climate system – without an increase in GHG) with that of a TFRV. Note that our trend-free random variable (TFRV), it is devoid of trends not only in the mean value but in variance and other moments. In the parlance of probability this is an independent, identically distribution (I.I.D.) random variable. Thus, one expects the climate internal variability to affect both, record highs and lows and exceed I.I.D variance. Yet, the PI control run samples are much closer to the expected results of TFRV than the observational data set.

Histograms of rho and of the high and low number of observed records over the TFRV expected values (NR/Exp) show that for the model's PI control realizations these are much closer to TFRV than to the observational SST dataset (ERSST-v5). While for the observations (Fig. 2 of the original manuscript), the peaks of the distribution the number of high (low) records was clearly above (below) the expected TFRV value, for the PI control run samples, the peak of the NR/Exp distributions is at 1 (Figure S1 below). The comparison of rho histograms of confirm this result. For the observations, the distribution of rho peaks much higher than 0 (Fig. 2 of the original manuscript), while in the Figure S1 below rho peaks at 0.

Recall that the internal variability *alone* is expected to increase the number of record highs and record lows evenly. However, we do not see this. The expected standard deviation of the records for TFRV is 1.8, and the model's PI control realizations have a standard deviation of 1.9. This resemblance of the records drawn from the GCM PI control with the TFRV reaffirms robustness of record-breaking in this novel context.

[Figure]

Fig S1: Histograms of observed over expected number of records and rho for the PI control run samples of the MRI-ESM2-0 CMIP6 global climate model.

The similarity between the number of records of TFRV and the PI control run samples, in contrast to the distribution of the observed number of records, is confirmed in the boxplots of Figure S2 below. (Readers may recall that boxplots allow comparison of distributions at a glance. The colored box is delimited by the first and third quartiles, Q1=P25 and Q3=P75. The thick line inside the box shows the median for each distribution. The size of the whiskers is 1.5 the interquartile range. Therefore, any point outside the whiskers is considered an outlier.)

Figure S2 also shows boxplots of the modelled SST NR/Exp of the PI control run samples and the projected SST NR/Exp for 2015-2100 (using the SSP2-4.5 scenario), towards addressing question 2 of the reviewer 2.

[Figure]

Fig S2: Boxplots of the expected number of high and low records and of rho for the observed SST between 1946-2020, TRFV, modelled SST of the PI control run samples, modelled SST between 1946-2020, projected SST between 2015-2100 (using the SSP2-4.5 scenario). The simulations were from the MRI-ESM2-0 CMIP6 global climate model.

Figure S3 shows the mean spatial distribution of the high and low NR/Exp and rho calculated from 1000 maps of the number of records (each of them for a random 75-year sample) of the modelled PI control run samples. We observe that the spatial pattern is almost homogeneous with weak values of NR/Exp and rho (close to 1 and 0, respectively). Unlike the observational results, there are no robust cooling or warming islands in these maps. The cooling and warming regions are due to sampling fluctuations, in contrast to our observational results

**a) MRI–ESM2–0 PI (no GHG) RHigh/Exp**

[Figure]

**b) MRI–ESM2–0 PI (no GHG) RLow/Exp**

[Figure]

**c) MRI–ESM2–0 (no GHG) $\rho = \ln(R_{High}/R_{Low})$**

Figure S3: Mean spatial distribution of the high and low NR/Exp and rho calculated from 1000 maps of the number of records (each of them for a random 75-year sample) of the modelled MRI-ESM2-0 PI control run samples.

The Kolmogorov-Smirnov (KS) test shows the following statistics for the MRI-ESM2-0 PI-control run samples:

Dhigh = 0.041
Dlow = 0.035

The statistics of the test indicate that the number of records of the MRI-ESM2-0 PI control run samples are significantly different from those expected for a TFRV distribution, both for RHigh and RLow. However, for the observations in the manuscript we got much higher D values, around 0.3 (far off the scale of the figure).

[Figure]

Figure S4: Cumulative frequency distributions of the high and low number of records for the MRI-ESM2-0 PI-control run samples compared to the TRFV distribution (a and b). Figure c shows the results of the statistics D of the Kolmogorov-Smirnov (KS) test for PI-control run samples compared to distribution of D we would get from a TFRV Monte-Carlo distribution.

We also compared the rho distribution from the modelled PI control run samples with the values of rho we got from Monte-Carlo simulations for TFRV. We see that the difference between these two distributions is very small. The statistics of the KS test for these distributions was Drho=0.015.

[Figure]

Figure S4: Comparison between the frequency distribution of rho for the MRI-ESM2-0 PI-control run samples and Montecarlo simulations of a TRFV.

The distribution of the deviation from expectancy (in standard deviation units) of the PI control run samples also peaks at 0 (Figure S5), unlike our results for the observational dataset, which were clearly shifted to positive values in the case of high records and negative values for low records (Figure 5 of the original manuscript).

[Figure]

Figure S5: Histograms of the high and low deviation from expectancy of the number of records for PI control run samples the MRI-ESM2-0 CMIP6 global climate model.

Boxplots show that the deviation from the expectancy of the PI control run samples is very similar to that of a TRFV (Figure S6). On the other hand, the deviation from expectacy for the observed, modelled and projected SSTs number of records are quite different from that of a TFRV, corroborating the results shown in Figure S2.

[Figure]

[Figure]

Figure S6: Boxplots of the high and low deviation from expectancy of the number of records for the: observed SST between 1946-2020, TRFV, modelled SST of the PI control run samples, modelled SST between 1946-2020, projected SST between 2015-2100 (using the SSP2-4.5 scenario). The simulations were from the MRI-ESM2-0 CMIP6 global climate model.

"2. The results may depend to a large extent on the sample size. As shown in Equation 2, the broken k-record varies with the length of the time series. In other words, the results may be sensitive to the length of the time series and not robust."

As argued above, record-breaking statistics is less sensitive to the time series length than other methods. We revised the manuscript accordingly. Specifically, we emphasize that, as the probability of breaking a record decreases monotonically at a rate of 1/n, the expected number of records in a TFRV (Equation 2), increases logarithmically, the slowest possible convergence.

**References**

Meehl, Gerald A., Catherine A. Senior, Veronika Eyring, Gregory Flato, Jean-Francois Lamarque, Ronald J. Stouffer, Karl E. Taylor, and Manuel Schlund. "Context for interpreting equilibrium climate sensitivity and transient climate response from the CMIP6 Earth system models." *Science Advances* 6, no. 26 (2020): eaba1981.

Zelinka, Mark D., Timothy A. Myers, Daniel T. McCoy, Stephen Po-Chedley, Peter M. Caldwell, Paulo Ceppi, Stephen A. Klein, and Karl E. Taylor. "Causes of higher climate sensitivity in CMIP6 models." *Geophysical Research Letters* 47, no. 1 (2020): e2019GL085782.

---

## Author Comment (AC2)

**Record-breaking statistics detect islands of cooling in a sea of warming**

Elisa T. Sena, Ilan Koren, Orit Altaratz, and Alexander B. Kostinski

Submitted to Atmospheric Chemistry and Physics

**Answers to Reviewer #2**

The authors explore the SST trends over the last 75 years and discuss the spatial characteristics. They use a trend-free random variable technique along with standard methods to test statistical significance. The number of high records clearly outweighs the number of low records, with the majority of the SST grid points showing a warming trend. While I consider the paper sound from a methodological point of view, I'm not sure whether ACP is the ideal outlet given its statistical nature. The results do merit publication, but in order to make it work for ACP I would suggest incorporating CMIP6 models to add more analytical (i.e. physical) content to the discussion of the observed trend or their differences for that matter. I therefore suggest major revisions to make it fit for ACP.

**Answer:** We thank the reviewer for the comments and suggestions and we are glad to see that "the results do merit publication". We are happy to revise the paper along the lines recommended by the reviewer and detailed below.

In the original version we aimed to show how measurement-based SST trends are reflected in the record breaking statistics (RBS). We had three main objectives:

1) to introduce an RBS analysis for maps, where the spatial distribution of RBs conveys important information for areas that are more susceptible to changes as well as anomalous areas resistant to change (RB differences are of opposite sign);

2) to show how robust and straightforward RBS is: we offer a direct and simple measure for the significance of deviation from non-trend data in two scales: we can estimate it per-pixel and we can estimate significance for a cluster of pixels; and

3) to quantify the severity of climate change using a new measure on canonical, exhaustively tested digital data.

The reviewers comments helped us to clarify the above points. Even though the focus of the paper is on observations, we now realize that RBS can be of great use in examining performance of global climate models (GCM) and, in particular, infer the contribution of the internal climate variability to SST trends and to future projections.

As suggested by the reviewer, we performed a thorough analysis aiming to compare SST trends from observations, TFRV, GCM pre-industrial (PI) GHG runs, GCM during the study period and future GCM projections. The main conclusions were: i) the distribution of records from GCM PI control run samples is very similar to the one from a TFRV, ii) the GCM is able to reproduce the observed records, and iii) RBS application to future GCM projections shows remarkable SST trends, with even more frequent high RBs and less frequent low RBs.

We revised the manuscript to include these results (Figures S2 and S6 below) and conclusions from this analysis. Remaining figures from this analysis may be placed in the Supplement. We also applied RBS to assess trends in an alternative observational dataset and we applied ENSO correction to the original dataset to verify the robustness of RBS. We discuss the results of the correction and the comparison with another observational data set in the revised manuscript and we included the figures emerging from this analysis in the Supplement.

Overall, I have three suggestions which may help to improve the manuscript:

1 Use CMIP6 pre-industrial control runs (random 75 year chunks) in order to compare the results when applying the TFRV technique. In order to obtain meaningful results (or learn something), I suggest selecting a dedicated CMIP6 subsample with a transient climate response (TCR) similar to observations, i.e. 1.3-1.8K. This way models with unrealistically high internal variability and/or too rapid southern ocean warming are likely omitted, which helps to confine the 'true' range of low-frequency internal variability. Taking all CMIP6 at face value without subsampling won't help us to distill the magnitude of the unforced trends.

**Answer:** Thanks for the suggestion. Following the reviewer's comments we have analyzed MRI-ESM2-0 CMIP6 global climate model (GCM). This model was chosen because, according to Meehl et al., 2020, it has a transient climate response (TCR) of 1.6 K (midrange of the reviewer), and because it predicts moderate increases in temperature, relative to other CMIP6 models and presents low residues (Zelinka et al., 2020 – supplementary material).

Record breaking statistics was applied to global time series of simulated SST (1 x 1 degree lat/lon) in the **pre-industrial control run** (PI control) from the model, which keeps the GHG levels unchanged since 1850 (considered as the reference year for the transition from the pre-industrial era to the present). 701 years of PI control run were available for this model. As suggested, for each grid cell we took 1000 random 75-year chunks from the PI control global simulation of MRI-ESM2-0 model to explore the effect of natural variability in the number of records. Ties were dealt with by adding a small random noise to each SST value.

Several analyses were done aiming to compare the results related to the number of records of the modelled PI control (which accounts only for internal variability of the climate system – without an increase in GHG) with that of a TFRV. Note that although our reference is named a trend-free random variable (TFRV), it is devoid of trends not only in the mean value but in variance and other moments as well. In other words, it is an independent, identically distribution (I.I.D.) random variable. Thus, we expect the climate internal variability to affect record highs and lows and exceed I.I.D variance. We see that the PI control run samples, are significantly closer to the expected results of TFRV compared to the results we obtained from the observational data set used in the original manuscript.

Histograms of rho and of the high and low observed number of records over the TFRV expected values (NR/Exp) show that for the model's PI control realizations these metrics are much closer to that of TFRV than the results we got from the observational SST dataset (ERSST-v5). While for the observations (Fig. 2 of the original manuscript), the peaks of the distribution the number of high (low) records was clearly above (below) the expected TFRV value, for the PI control run samples, the peak of the NR/Exp distributions is at 1 (Figure S1 below). The comparison between the histograms of rho also confirm this result. For the observations, the distribution of rho peaks much higher than 0 (Fig. 2 of the original manuscript), while in the Figure S1 below rho peaks at 0.

Recall that the internal variability alone is expected to increase the number of record highs and record lows evenly. Yet, the expected standard deviation of the records of a TFRV is 1.8, and the model's PI control realizations have a standard deviation of 1.9. This resemblence of the records distribution drawn from the GCM PI control with the TFRV gives us more confidence in appling record-breaking statistics to verify trends climate time series and, once again, confirms the robustness of this method. We thank the reviewer for their suggestion to use a model with transient climate response (TCR) similar to observations, since it has likely suppressed the variability of records due to unrealistic representations of the internal variability.

[Figure]

Fig S1: Histograms of observed over expected number of records and rho for the PI control run samples of the MRI-ESM2-0 CMIP6 global climate model.

The similarity between the number of records of a TFRV and the PI control run samples, in contrast to the distribution of the observed number of records, is confirmed in the boxplots of Figure S2 below. (Readers may recall that boxplots allow comparison of distributions at a glance. The colored box is delimited by the first and third quartiles, Q1=P25 and Q3=P75. The thick line inside the box shows the median for each distribution. The size of the whiskers is 1.5 the interquartile range. Therefore, any point outside the whiskers is considered an outlier.)

Figure S2 also shows boxplots of the modelled SST NR/Exp of the PI control run samples and the projected SST NR/Exp for 2015-2100 (using the SSP2-4.5 scenario), towards addressing question 2 of the reviewer. The observed, modelled and projected NR/Exp are quite different from that of a TFRV, clearly shifted to higher values in case of high records. For the low records, the distribution is shifted towards negative values in the observed and projected scenario. Note that the GCM could capture the general trend revealed by record-breaking statistics, but smoothen its magnitude. Future projections, however, are outstanding and will be further discussed in question 2.

[Figure]

Fig S2: Boxplots of the expected number of high and low records and of rho for the observed SST between 1946-2020, TRFV, modelled SST of the PI control run samples, modelled SST between 1946-2020, projected SST between 2015-2100 (using the SSP2-4.5 scenario). The simulations were from the MRI-ESM2-0 CMIP6 global climate model.

Figure S3 shows the mean spatial distribution of the high and low NR/Exp and rho calculated from 1000 maps of the number of records (each of them for a random 75-year sample) of the modelled PI control run samples. We observe that the spatial pattern is almost homogeneous with weak values of NR/Exp and rho (close to 1 and 0, respectively). Unlike the observational results, there are no robust cooling or warming islands in these maps. The cooling and warming regions are due to sampling fluctuations, in contrast to our observational results

**a) MRI–ESM2–0 PI (no GHG) RHigh/Exp**

[Figure]

**b) MRI–ESM2–0 PI (no GHG) RLow/Exp**

[Figure]

**c) MRI–ESM2–0 (no GHG) $\rho = \ln(R_{High}/R_{Low})$**

Figure S3: Mean spatial distribution of the high and low NR/Exp and rho calculated from 1000 maps of the number of records (each of them for a random 75-year sample) of the modelled MRI-ESM2-0 PI control run samples.

The Kolmogorov-Smirnov (KS) test shows the following statistics for the MRI-ESM2-0 PI-control run samples:

Dhigh = 0.041
Dlow = 0.035

The statistics of the test indicate that the number of records of the MRI-ESM2-0 PI control run samples are significantly different from those expected for a TFRV distribution, both for RHigh and RLow. However, for the observations in the manuscript we got much higher D values, around 0.3 (far off the scale of the figure).

[Figure]

Figure S4: Cumulative frequency distributions of the high and low number of records for the MRI-ESM2-0 PI-control run samples compared to the TRFV distribution (a and b). Figure c shows the results of the statistics D of the Kolmogorov-Smirnov (KS) test for PI-control run samples compared to distribution of D we would get from a TFRV Monte-Carlo distribution.

We also compared the rho distribution from the modelled PI control run samples with the values of rho we got from Monte-Carlo simulations for TFRV. We see that the difference between these two distributions is very small. The statistics of the KS test for these distributions was Drho=0.015.

[Figure]

Figure S4: Comparison between the frequency distribution of rho for the MRI-ESM2-0 PI-control run samples and Montecarlo simulations of a TRFV.

The distribution of the deviation from expectancy (in standard deviation units) of the PI control run samples also peaks at 0 (Figure S5), unlike our results for the observational dataset, which were clearly shifted to positive values in the case of high records and negative values for low records (Figure 5 of the original manuscript).

[Figure]

Figure S5: Histograms of the high and low deviation from expectancy of the number of records for PI control run samples the MRI-ESM2-0 CMIP6 global climate model.

Boxplots show that the deviation from the expectancy of the PI control run samples is very similar to that of a TRFV (Figure S6). On the other hand, the deviation from expectacy for the observed, modelled and projected SSTs number of records are quite different from that of a TFRV, corroborating the results shown in Figure S2.

[Figure]

[Figure]

Figure S6: Boxplots of the high and low deviation from expectancy of the number of records for the: observed SST between 1946-2020, TRFV, modelled SST of the PI control run samples, modelled SST between 1946-2020, projected SST between 2015-2100 (using the SSP2-4.5 scenario). The simulations were from the MRI-ESM2-0 CMIP6 global climate model.

 2 I suggest expanding the analysis to other SST datasets such as HadSST4, Kadow et al. 2020 (https://www.nature.com/articles/s41561-020-0582-5.pdf) or the (preliminary) version of HadISST2. They do tend to have small difference even after 1946 (between 1950-1980 in particular), hence it would be insightful to test them separately with the same method. It will certainly increase the robustness of the results. Expanding the analysis further by using the same subsample of CMIP6 SSP2-4.5 simulations between 1946-2020 would have the potential to add further insight into the forced nature of the signal and the physical reasons for missing warming trends in some ocean regions.

**Answer:** We really appreciate the suggestions but HadSST4 is not interpolated and therefore is not suitable for applying record breaking statistics. The interpolated data set, HadISST2 is not

available for the general public yet. These are some of the reasons that lead us to choose ERSST-v5 as the primary data set used in revising this work as recommended by the reviewer. We have now used a previous version of the suggested data set, HadISST, to verify the consistency of our results. Overall, record-breaking statistics point to similar conclusions for either of the two data sets. As ERSST-v5, HadISST also presents an asymmetry in the number of records, with more high than low records (Figure S7). However, for HadISST the number of high records that surpass the TFRV expected value is 96% (as opposed to 83% for the ERSST-v5 data set). Furthermore, HadISST shows weaker rho values than ERSST-v5 results. There are some features worth mentioning in the spatial pattern as well, such as the cooling spots in the North Pacific and North Atlantic, that also showed up in Figure 2 of the original manuscript. The cold spot in the Southern Ocean is not evident in this data set, though.

[Figure]

Fig S7: Spatial distribution and histograms of observed over expected number of records and rho for the observational HadISST data set during the study period.

As suggested by the reviewer, we've expanded the analysis further and applied record breaking statistics to the MRI-ESM2-0 simulations during the same time period as used in our observational analysis (1946-2020). For that, a combination of the historical data set and scenario projection 2 (SSP2-4.5) was used. The spatial distribution and histograms that resulted from these analyses are shown in Figure S8. They show that the excess of records highs over the TFRV expected value is similar to observations, while the distribution of the low NR/Exp peaks around 1, resulting in positive but smaller values of rho relative we to the observational results, shown in Figure 2. The boxplots shown in Figure S2 help to interpret these results. Note the resemblance of the spatial patterns of this global model simulation (Figure S8) and the observation-based results (Figure 2 of the manuscript). There are some cooling regions in North Pacific and North Atlantic and a cooling spot in the Southern Ocean, just like the interesting features that we spotted in the results we got from the observational data set (Figure 2). There is also cooling in the Indian Ocean, between Africa and Australia and the east coast of South America. These two features also appear in Figure 2, although not so prominently as the previous ones.

[Figure]

[Figure]

[Figure]

[Figure]

[Figure]

[Figure]

Fig S8: Spatial distribution and histograms of observed over expected number of records and rho for the SST of the MRI-ESM2-0 CMIP6 global climate model during the study period (1946-2020).

We also aplied record breaking statistics on the scenario projection SSP-2 4.5 of MRI-ESM2-0 CMIP6 model, between 2015 to 2100. The results show that, according to this model, in the future the high record breaking will be even more frequent than in the present, while low record breaking will be even less frequent (Figures S9 and S2). At least two cold spots are noticeable: a large one south of Greenland (presently known as the cold blob, already discussed in the original manuscript), and a second one in the Southern Ocean, southeast of Australia, which could be related to the cold spot we discovered in the historical observational data set ERSST-v5.

[Figure]

Fig S9: Maps and histograms of observed over expected number of records and rho for the SST projections (SSP2-4.5) of the MRI-ESM2-0 CMIP6 global climate model from 2015-2100.

3 It might be worthwhile applying an ENSO correction to the data. The method provided in Foster and Rahmstorf 2011 (https://iopscience.iop.org/) using multiple regression to filter out all natural

factors (volcanoes, solar variability and ENSO), could offer some more interesting insights, both as far as trends as well as high and low record statistics at grid point level are concerned.

**Answer:** We did the correction suggested by the reviewer and the results barely changed. This robustness of the record breaking statistics with respect climate time series trends in the mean, despite the internal variability of the system.

To quantify how the internal variability of the system influences the number of records, a correction was applied to the dataset to remove the contribution of natural factors (ENSO, volcanic and solar signals) from SST. For that, a multiple linear regression of globally-averaged SST anomaly against a linear term and lagged time series of standardized SOI (https://psl.noaa.gov/gcos_wgsp/Timeseries/SOI/), Tau (Sato et al., 1993 - https://data.giss.nasa.gov/modelforce/strataer/) and TSI indexes (https://www.ncei.noaa.gov/data/total-solar-irradiance/access/monthly/) was performed for the study period (Foster and Rahmstorf 2011, Lean and Rind, 2008). Table S1 shows the time lags (in months) used and the resulting coefficients for each term.

Table S1: Lags and coefficients from the multiple linear regression of SST anomaly by the several indexes.

|              | Lag (months) | Coefficients      |
|--------------|--------------|-------------------|
| Linear term  | -            | $0.0089 \pm 0.0001$ |
| TSI          | 0            | $0.006 \pm 0.002$   |
| SOI          | 3            | $-0.035 \pm 0.002$  |
| Tau          | 11           | $-0.030 \pm 0.002$  |

For each pixel, the contribution of each natural forcing was removed from the SST. Figures S10 and S11 show, respectively, the time series of the mean global SST anomaly, and maps of high and low records before and after removing the contribution of natural factors. It is evident from these figures that the number of records remained practically invariant to this adjustment. This shows that the record breaking statistics is very robust and can be used to find trends in climate time series, despite the internal variability of the system.

[Figure]

Figure S10: Time series of the mean global SST anomaly before and after the removal of the contribution of natural internal variability.

[Figure]

Figure S11: Spatial distribution of the number of high and low records before and after the removal of the contribution of natural internal variability.

**References**

Meehl, Gerald A., Catherine A. Senior, Veronika Eyring, Gregory Flato, Jean-Francois Lamarque, Ronald J. Stouffer, Karl E. Taylor, and Manuel Schlund. "Context for interpreting equilibrium climate sensitivity and transient climate response from the CMIP6 Earth system models." *Science Advances* 6, no. 26 (2020): eaba1981.

Sato, M., J.E. Hansen, M.P. McCormick, and J.B. Pollack 1993: Stratospheric aerosol optical depth, 1850-1990. *J. Geophys. Res.* **98**, 22987-22994.

Zelinka, Mark D., Timothy A. Myers, Daniel T. McCoy, Stephen Po-Chedley, Peter M. Caldwell, Paulo Ceppi, Stephen A. Klein, and Karl E. Taylor. "Causes of higher climate sensitivity in CMIP6 models." *Geophysical Research Letters* 47, no. 1 (2020): e2019GL085782.

---

## Author Response (AR2)

**Record-breaking statistics detect islands of cooling in a sea of warming**

We thank the editor the important comments and for the constructive and positive review process. We have improved all the figures' quality, clarify and unify the labeling, as well as reduced the space between them. We had to leave the figures title to avoid confusion between the different maps and measurements. We hope that this is acceptable.

We note that the figures quality (as separate files) is much better than how they appear in the word document. We will be happy to upload the separate figures' files for the final version.

Many thanks and best wishes,
Ilan Koren

---

## Author Response (AR3)

5        **Record-breaking statistics detect islands of cooling in a sea of warming**

We thank the editor the important comments. We have corrected the references and provided all the needed information. Since we did not find a section to upload the paper with track-changes, to show our changes, it is

10    attached below.

Many thanks and best wishes,
Ilan Koren

[revised manuscript text omitted]